# Adsorption of Fluoride onto Acid-Modified Low-Cost Pyrolusite Ore: Adsorption Characteristics and Efficiencies

**DOI:** 10.3390/ijerph192417103

**Published:** 2022-12-19

**Authors:** Phacharapol Induvesa, Radamanee Rattanakom, Sornsiri Sriboonnak, Chayakorn Pumas, Kritsana Duangjan, Pharkphum Rakruam, Saoharit Nitayavardhana, Prattakorn Sittisom, Aunnop Wongrueng

**Affiliations:** 1Bodhivijjalaya College, Srinakharinwirot University, Nakhon Nayok 26120, Thailand; 2Department of Environmental Engineering, Faculty of Engineering, Chiang Mai University, Chiang Mai 50200, Thailand; 3Research Center in Bioresources for Agriculture, Industry and Medicine, Chiang Mai University, Chiang Mai 50200, Thailand; 4Department of Biology, Faculty of Science, Chiang Mai University, Chiang Mai 50200, Thailand; 5Science and Technology Research Institute, Chiang Mai University, Chiang Mai 50200, Thailand

**Keywords:** acid-modification, adsorption, fluoride, groundwater, pyrolusite ore

## Abstract

Fluoride concentrations in the groundwater continue to be a major cause for concern in Thailand, particularly in the country’s north and west. The process of removing fluoride through adsorption has captured the attention of the abundance of ore in the mining industry. For the purpose of this investigation, the utilization of the adsorbent pyrolusite, which is a manganese mineral largely composed of MnO_2_, was a major component. Lab-scale experiments were conducted to investigate the efficacy of original pyrolusite ore (PA-1) and acid-modification PA (PA-2) created as low-cost adsorption materials for fluoride removal. The results of the adsorption rate in both PAs showed a fast rate of adsorption within 60 min of reaching equilibrium. According to the results of the adsorption capacity (q_e_) tests, PA that had been treated with an acid solution (PA-2) had the capacity to contain more fluoride (q_e =_ 0.58 mg/g) than the PA that had been used initially (PA-1) (q_e_ = 0.11 mg/g). According to the findings of an isotherm, primary adsorption behavior is determined by the effect that surface components and chemical composition have on porous materials. This is the first current study that provides a comparison between pyrolusite from Thailand’s mining industry and basic modified pyrolusite regarding their ability to remove a fluoride contaminant in synthetic groundwater by an adsorption process. Such an approach will be able to be used in the future to protect the community from excessive fluoride concentrations in household and drinking water treatment technology.

## 1. Introduction

The problem of an excessive concentration of fluoride in the groundwater in the northern part of Thailand is still a significant cause of concern for human wellbeing due to the hydrogeological profile of the region [1]. The range of fluoride concentration in the north, especially in the Chiang Mai and Lamphun provinces, is 0.1–16 mg [2,3]. Even though fluoride is beneficial for oral health and helps prevent dental cavities, consuming too much of it (approximately higher than 1.5 mg/L) can lead to dental fluorosis as well as other adverse effects [4,5]. Excessive fluoride exposure during dental organ development would result in the mineralization deficiency that causes dental fluorosis. The key period for the development of dental fluorosis in permanent teeth is from birth to age eight [4].

The inhabitants of the northern region of Thailand have been negatively impacted by the issue of excessive levels of fluoride pollution in the groundwaters of their region, which are used for drinking and utilized in the kitchen. Generally, in groundwater and natural water sources, the World Health Organization (WHO), the Environmental Protection Agency (EPA) of the United States, and the Food and Drug Administration of Thailand, all agree that 0.70 mg/L of fluoride is the minimum level that should be present in drinking water to ensure public health and safety [6]. The fluoride problem in groundwater generates a never-ending stream of challenges that must be surmounted. It is necessary to have both an awareness of fluoride’s dispersion and a comprehensive grasp of its many probable sources and especially removal technologies for management to be effective. Various water and groundwater treatment technologies have been applied to remove fluoride from water sources, such as electrodialysis [7,8], membrane filtration [9,10,11], and adsorption processes [12,13,14,15,16]. Of these technologies, when applied in a local community, adsorption is considered an excellent choice for defluorination due to its low cost, ease of design and operation, and high level of effectiveness [17].

One of the most important terms in the adsorption process is ‘adsorbent’. Many studies have reported the use of complex adsorbents to remove fluoride, such as magnetic nanocomposite [18], zeolite modification [19], bone char [20,21], ceramic [22], and magnetic carbon nanotube [23], although achieving an adsorbent requires several synthesis processes.

Consequently, the utilization of alternative low-cost adsorbents is one of the greatest choices available to a local community in terms of defluorination. There have been a number of studies conducted on low-cost adsorbents for fluoride removal, such as biomass, low-grade or abundant ore from mining, and natural ore [15,24,25], but their use is still rare and requires additional information and research into the adsorption behaviors of low-cost materials that do not require any complicated synthesis.

In the northern region of Thailand, mining operations for pyrolusite ore (PA) can be found. The most common manganese mineral is pyrolusite (manganese dioxide), which contains 60–63% of manganese [26]. However, some is considered low-grade pyrolusite ore and cannot be commercialized. Previous studies have reported that pyrolusite ore enhances the adsorption capacity of an aqueous pollutant [27], hence the good opportunity to examine pyrolusite ore, which is a low-cost material, as a method for fluoride removal in this study. Some research has reported that pyrolusite ore was applied for fluoride and phosphate removal in wastewater with its surface modified by Lanthanum ion dropping [28]. Based on the main proportion of pyrolusite being manganese or manganese oxide, previous studies investigated fluoride removal via other metal oxides, i.e., MgO showed the interaction via the electrostatic interaction as follows (1) [29]:Mg-(OH)_2_ + F^−^ → Mg-F + 2OH^−^(1)

Since the adsorbent used in this work also belongs to the group of metal oxides, pyrolusite mainly consists of MnO_2_, it will be able to represent the same specific interaction in most probable Equations (2) and (3) [30,31].
MnO_2_∙yH_2_O → Mn2O_3_∙xH_2_O → Mn(OH)_2_(2)
Mn-(OH)_2_ + F^−^ → Mn-F + 2OH^−^(3)

However, the abundance of OH radicals in the solution might be decreased in fluoride adsorption, with further surface modification being necessary for enhancing its capacity. One of the most basic and low-cost modifications is acid-modification. In modifying the surface of metal oxide ore, some acid-modification with manganese on the surface of zeolite has been shown to have a good adsorption efficiency for fluoride removal [19]. The acid surface modification of pyrolusite may improve the positive surface charge and enhance the negative ion, such as fluoride [32]. Therefore, the combination of abundant pyrolusite ore and the ability to modify its surface with an acidic solution might be a novel option for defluorination studies.

The purpose of this study was to investigate the use of abundant pyrolusite ore and compare it with acid surface modification for fluoride removal. The novelty of this study is its aim to establish a simple materials and modification method that can be applied to excessive fluoride to protect human health, as well as being the first investigation of low-cost pyrolusite abundant ore in the northern mining industry in Thailand to remove excess fluoride from groundwater and improve drinking water usability. The adsorption efficiency and adsorption behavior of prepared absorbents were evaluated via adsorption kinetics and isotherms. Scanning electron microscopy (SEM) coupled with microchemical composition (EDS), porosity, Brunner–Emmet–Teller (BET), and FTIR were used to characterize the physical and chemical properties of pyrolusite and its modified surface. Additionally, the effect of stability on the adsorbents was investigated in this study.

## 2. Method and Methodology

### 2.1. Materials and Chemicals

Sodium fluoride (NaF, 99.99%) was purchased from Merck. Hydrochloric acid (HCl, 37%) and Sodium hydroxide (NaOH, 99%) were purchased from RCI Lab Scan. Zirconyl chloride octahydrate (ZrOCl_2_, 8H_2_O) and SPADNS reagents (C_16_H_9_N_2_Na_3_O_11_S_3_) were purchased from Merck. All were either analytical or reagent grade.

### 2.2. Preparation of Pyrolusite Ore Adsorbent (PA)

To produce the pyrolusite ore adsorbent (PA), pyrolusite ore was obtained from a pyrolusite mining area in Chiang Mai province, Thailand. Then, it was processed as follows: The pyrolusite ore was crushed into a small size of 1–2 mm. After that, the PA samples were divided into two types including a PA cleaned and soaked with deionized water (DI), called PA-1, and a modified PA cleaned and soaked with 0.1 N HCl for 24 h and washed with DI water until the pH closed to 7.00, called PA-2 (acid-modification). Then, both prepared PAs were dried in an oven at 110 °C for 24 h before use.

The summarized details of PA preparation including original PA (PA-1) and acid-modification (PA-2) are shown in Table 1.

### 2.3. Characterization of PAs

The physical and chemical properties of materials were analyzed with several methods. The surface morphologies and microchemical compositions of PA-1 and PA-2 were examined using a SEM/EDS (JSM-IT300), JEOL Ltd., Tokyo, Japan. The porosity and surface area (BET) were measured by the nitrogen adsorption isotherm at 77 K on a Micromeritics 3Flex adsorption analyzer, Micromeritics Instrument Corporation, USA. The surface functional group was measured using a FT-IR spectrometer, INVENIO, in a range of 400–4000 cm^−1^, BRUKER, Germany. The points of zero charge (pH_PZC_) of PA-1 and PA-2 were adapted from [33] and determined as follows: PA-1 and PA-2 were weighed for 1.0 g and put in 200 mL of deionized water at a pH range of 1.0 to 14.0, adjusted by nitric acid or sodium hydroxide. Then, the mixed samples were shaken by using a shaker (GFL, Orbital shaker 3017, Greater Hanover, Germany) at 200 rpm for 24 h. After that, final pH of each sample was measured. The initial and final pH values were plotted to obtain the point of zero charge (pH_PZC_) at the crossing point between the lines connecting the pH data and the diagonals connecting the equal initial and final pH.

### 2.4. Adsorption Kinetics and Isotherm of the PAs

Synthetic groundwater with a fluoride concentration of approximately 10 mg/L was used in the batch adsorption experiments with reference to the fluoride concentration of groundwater in an actual contaminated site in Chiang Mai Province, Thailand. The synthetic water was prepared by dissolving sodium fluoride (NaF) in deionized water. All kinetics and isotherms adsorption experiments were performed in duplicates.

The adsorption kinetics experiments were conducted by mixing 1.5 g of the PAs into 100 mL of synthetic groundwater at pH 7.0 (controlled by a phosphate buffer). These samples were mixed by a rotary shaker (GFL, Orbital shaker 3017, Greater Hanover, Germany) at 200 rpm at 25 °C. The mixing was stopped at different adsorption times ranging from 0 to 24 h, and PA-1 and PA-2 were separated from the solution by using a nylon syringe filter (dia. 13 mm, nominal pore size 0.45 µm, Chrom Tech). The filtrates were analyzed for residual fluoride concentration by using the colorimetric method at a wavelength of 570 nm (Jenway 6400 Spectrophotometer, Jenway, London, UK).

The adsorption isotherms were studied by varying the initial fluoride concentrations from 2 mg/L to 10 mg/L with the same solid and liquid ratio as reported in the kinetic study. The pH of the solution was controlled by a phosphate buffer at 7.00. The equilibrium time was based on the kinetics study results. After adsorption at the equilibrium time, the solution was filtrated and residual fluoride was measured as described in the kinetics study.

## 3. Results and Discussion

### 3.1. Physio-Chemical Properties of Materials

Morphological characteristics were initially used to investigate the physical properties of materials. The morphology of prepared materials is illustrated in Figure 1. Similar to other studies [28], PA-1 (washed with deionized water) had rough surfaces. However, the particle morphology of HCl modification (PA-2) was different from PA-1, leading to an irregular surface and porosity. The shape and appearance showed a rough surface and higher porosity than PA-1 due to HCl modification, which could be used to remove surface impurities and enhance the physical properties of materials. Therefore, fluoride adsorption into the pore spaces increased after the materials were acid-modified [34].

The quantitative chemical compositions of PA-1 and PA-2 were analyzed using an energy-dispersive X-ray microanalyzer (SEM-EDS). The chemical composition of PA-1 exhibited the types of elements shown in Table 2. The highest proportion (% wt) of PA-1 was Mn, followed by Fe, O, C, N, Al, and Si, respectively. The elements in the prepared samples showed a similar composition to that of the original pyrolusite, which contained the complex of MnO_2_, CaO, Fe_2_O_3_, SiO_2_, Al_2_O_3,_ and MgO [27]. In addition, the PA-2 (washed with 0.1 N HCl) showed a similar trend to the PA-1, however, some slight changes occurred. The Si and N disappeared, and some mineral content such as Al was reduced after acidic modification [35].

Appendix Ashows the results of surface functional group characterization using FTIR of the original PA (PA-1) and acid-modified-PA (PA-2). FITR analysis was investigated in a range of 400–4000 cm^−1^. A pyrolusite peak (MnO_2_) was exhibited in both PA-1 and PA-2 wavenumber brands at 650–800 cm^−1^ [32]. The surface changes of acid-modified-PA (PA-2) were observed as a new brand between 1700–1800 cm^−1^ and 2900 cm^−1^, indicating that the C=O and C-H stretching vibrations of hydroxyl groups are associated with hydrochloric acid molecules via hydrogen bonding [36,37]. As a result, the success of acidic modification could be demonstrated. 

The porosity and surface area of both PAs were investigated by the nitrogen adsorption isotherm (Figure 2 and Table 3) via the BET and BJH methods, respectively. The calculated BET surface areas of PA-1 (washed with DI water) and PA-2 (washed with acid) were 111.48 and 108.93 m^2^/g, respectively. The pore sizes of PA-1 and PA-2 materials, measured by Barret–Joyner–Halenda (BJH) methods, were 3.65 and 4.62 nm, respectively. PA-2 had larger pores than PA-1 because it had been activated with an acid solution to improve its porous properties. In addition, the pore volumes of PA-1 and PA-2 were 0.112 and 0.107 cc/g, respectively. The pore volume of both adsorbents showed a small reduction in PA-2 compared with PA-1, which indicated the possession of PA-1 by residual HCl molecules. Furthermore, the hysteresis loops of PA-1 and PA-2 represent mesopore adsorbents with pore sizes ranging from 2 to 50 nm. The zeta potentials of PA-1 and PA-2 at varying ranges of pH are shown in Appendix A. At the point of zero charge (pH_pzc_), the pH of both materials was approximately 4.00.

### 3.2. Adsorption Kinetics Study

The adsorption kinetics studied of fluoride on PA-1 and PA-2 materials were investigated to determine the rate of adsorption, as displayed in Figure 3. The adsorption rate reached equilibrium in all adsorbents after 60 min, indicating the potential for rapid fluoride removal due to the rapid rate at which an equilibrium state was reached between the adsorbate and the adsorbent, and the result followed a similar trend seen with other ore adsorbents [24]. An important and practical component of this rapid adsorption is the use of tiny reactor volumes, which guarantee great efficiency at a low cost [38]. Comparing PA-1 and PA-2, the adsorption capacity of PA-2 was higher than PA-1. This may be possible with a surface modification of 0.1 N HCl to enhance positive surface charge [35,39]. Although the BET results for PA-1, including surface area and pore volume, are higher than those for PA-2, the adsorption capacity for PA-1 is lower. This could confirm that the adsorption mechanism of pyrolusite ore on fluoride is not only a physical adsorption mechanism. Another result of the material characterization could confirm the adsorption mechanism and that is the pH_pzc_ of the materials. The point of zero charge for both adsorbents was 4.00; however, the pH of the solution was greater than 4.00, indicating that the adsorbent surface was negatively charged (pH > pH_pzc_), resulting in low adsorption capacities in both adsorbents [38]. This would confirm that the adsorption of prepared ore (PA) would make it possible to allow for a variety of types of adsorption mechanisms, such as physical, chemical, and the effect of electrostatic interaction between the material and the solution. If the surface of the material is modified to be positively charged, it will be suitable for use with a fluoride solution (a negatively charged solution) to enhance its adsorption capacity [34].

Two different kinetic models including pseudo-first order and pseudo-second order equations were applied to quantitatively determine the rate of defluorination. The equation of pseudo-first order and pseudo-second order models are given by Equations (4)–(6) [21,40,41], respectively.

The pseudo-first order equation is presented by the following equation:(4)$$\ln\left(\frac{q_{e}}{q_{e}-q_{t}}\right)=k_{p1}t$$

Which can be rearranged in linear form as follows:(5)$$\log\left(q_{e}-q_{t}\right)=\log(q_{e})-\frac{k_{p1}t}{2.303}$$
where *q_t_* is the adsorption capacity at time *t* (mg/g), *q_e_* is the adsorption capacity at equilibrium (mg/g), *t* is time (min), and *k_p_*_1_ is the pseudo-first-order rate constant (min^−1^).

The pseudo-second order equation is applied to explain the chemical reactions of heterogeneous materials. This model is presented by the following equation:(6)$$\frac{t}{\;q_{t}}=\frac{1}{k_{p2}q_{e}^{2}}+\frac{t}{q_{e}}$$
where *q_t_* is the adsorption capacity at time *t* (mg/g), *q_e_* is the adsorption capacity at equilibrium (mg/g), *t* is time (min), and *k*_*p*2_ is the pseudo-second order rate constant (g/mg∙min).

Based on the pseudo-second order equation, the initial rate of adsorption, h (mg/g/min) at *t* = 0 can be calculated using Equation (7)
(7)$$h=k_{2}q_{e}^{2}$$

All calculations of the adsorption kinetic models are summarized in Table 4. The kinetic data of PA-1 and PA-2 represented a good correlation (R^2^) with the pseudo-second order model (R^2^ = 0.9952 and 0.9957, respectively.). However, the pseudo-first order model was not well fitted with both materials (R^2^ less than 0.7). In addition, the adsorption capacity from the pseudo-second order model (q_e_,_cal_) was close to the experimental results (q_e_,_exp_). The results indicated that a good performance of both PAs and fluoride probably represented a combination of chemical sorption and a dependency on the effect of the active site of adsorbent and adsorbate, especially in the presence of surface modification with an acid solution [19]. When considering the K_2_ parameter from the pseudo-second order model, the rate of PA-2 was higher than that of PA-2 due to the effect of a flat surface resulting in an increased speed of fluoride adsorption [28].

### 3.3. Adsorption Isotherms

The adsorption isotherm of two PA materials (PA-1 and PA-2) on fluoride removal are shown in Figure 4. Adsorption studies of fluoride on PA-1 and PA-2 were conducted at pH 7 controlled by a phosphate buffer. The graphical plot of all obtained isotherm data exhibited a linear function of fluoride adsorption. From the results, the adsorption isotherm of PA-2 showed higher adsorption capacities due to higher pore volume and perhaps because of acid modification slightly increasing the positive charged and active site properties [42].

Four isotherm models (linear, Langmuir and Freundlich and Sips model) were used to investigate the adsorption behavior of fluoride on pyrolusite ore adsorbents (Table 5). [41,43]. The concentration range of 2–15 mg/L was used due to their occurrence with a solid to liquid ratio of 15 g/L.

The linear model equation is shown as Equation (8)
(8)$$q_{e}=K_{p}C_{e}$$
where $K_{p}\;$ is the linear partition and $C_{e}$ is the concentration at equilibrium of fluoride.

The Langmuir isotherm defines monolayer sorption onto the surface, with sorption locating only at some sites. There are no interactions between the molecules. The model equation is shown as Equations (9) and (10):
(9)$$q_{e}=\frac{q_{0}K_{F}C_{e}}{1+K_{F}C_{e}}$$

Which can be rearranged in the linear form as follows:(10)$$\frac{1}{q_{e}}=\frac{1}{K_{F}q_{0}}\frac{1}{C_{e}}+\frac{1}{q_{0}}$$
where *q*_0_ is the amount of fluoride adsorbed per unit weight of PA in forming a complete monolayer on the surface (mg/g), *q_e_* is the total amount of fluoride adsorbed per unit weight of PA at equilibrium (mg/g), *C_e_* is the concentration of the fluoride in the solution at equilibrium (mg/L), and *K_F_* is the constant related to the energy of sorption (L/mg).

The Freundlich isotherm can be applied with non-ideal sorption on heterogeneous surfaces and with multilayer sorption. The model is shown by the following Equations (11) and (12):(11)qe=KFCe1n

Which can be rearranged in the linear form as follows:(12)$$logq_{e}=\frac{1}{n}logC_{e}+logK_{F}$$
where *q_e_* is the total amount of fluoride adsorbed per unit weight of PA at equilibrium (mg/g), *C_e_* is the concentration of the fluoride in the solution at equilibrium (mg/L), *K_F_* is the Freundlich constant which can express the capacity of the adsorption process (L/g), and *n* is the Freundlich constant, which explains the concentration of adsorption (dimensionless).

The Sips isotherm model is a combination of the Langmuir and Freundlich isotherm models, which are appropriate for predicting adsorption on heterogeneous surfaces. The equation is shown as Equation (13):(13)$$q_{e}=\frac{q_{ms}{\left(K_{s}C_{e}\right)}^{n_{s}}}{1+{\left(K_{s}C_{e}\right)}^{n_{s}}}$$
where $n_{s}$ is the Sips isotherm exponent, *K_S_* (L/µg) is the Sips adsorption constant and $q_{ms}$ is the maximum adsorption capacity (µg/g).

Based on isotherm results, the adsorption mechanism could be explained by the mentioned isotherm model of fluoride adsorption by PA-1 and PA-2. The correlation rankings, in ascending order (R^2^ highest to lowest), of both PA-1 and PA-2 were Freundlich > Sips ≅ Linear > Langmuir. This could be an indication of monolayer adsorption on the surface of PAs materials [44]. The statement of the Freundlich isotherm model defines both the heterogeneity of the surface and the exponential distribution of the active sites. To confirm the performance results of adsorption, the value of 1/*n* from the fitted isotherm model was used for prediction. When 1/*n* is higher than zero and does not exceed 1, the adsorption process is favorable. As a result, this study found 1/*n* values of 0.51 and 0.56 for PA-1 and PA-2, respectively, confirming that the fluoride adsorption process on PAs performed well, particularly at acid-modification (PA-2) sites [44].

In addition, the research that has been done on the adsorption of fluoride by low-grade pyrolusite is limited and needs to be supplemented with additional research for further investigation. Even though neither the original nor the acid-modified ore had an especially high adsorption capacity, both types of ore are still suitable for use as non-commercial materials in the process of reducing the amount of fluoride that is present in groundwater in order to comply with the regulatory standard that is in place. When the adsorption capacity of prepared pyrolusite in this study was compared with that of low-cost or abundant materials on fluoride removal, the results of the comparison were shown in Table 6. These materials included iron ore, dolomite ore, and *Polygonum orientale* Linn. The adsorption capacity of all materials on fluoride adsorption is based on the Freundlich adsorption capacity. When comparing the same initial fluoride concentration (10 mg/L) with abundance ore, the results demonstrated that PA-1 and PA-2 have a higher fluoride adsorption capacity than that of dolomite ore; this may be due to the smaller pore volume of dolomite (0.0031 cc/g) than that of either PA-1 or PA-2, which were 0.112 and 0.107 cc/g, respectively. Although the adsorption capacity of iron ore showed the highest efficiency, the equilibrium reaction time was 120 min at pH 6, which was higher than PA-1 and PA-2’s (60 min) at pH 7 because of the difference in temperature and pH in the reaction. Furthermore, because the selected materials are abundant, the cost of all adsorbents is only in the surface modification of the materials, such as acid or metal oxide modification. When comparing the modification cost, for example, of *Polygonum orientale* Linn., modified with Al_2_(SO_4_)_3_ (PO-Al) and acid-modified pyrolusite (PA-2), the functionalization cost of PO-Al will be slightly higher than PA-2 due to the price of Al_2_(SO_4_)_3_ being higher than hydrochloric acid.

According to the findings, this sort of adsorbent allows the prepared pyrolusite ore, as well as acid-modified materials, to be compatible with other types of materials. Even though some materials, such as iron ore, demonstrated good performance [24], pyrolusite is an abundant ore option from the northern region of Thailand that will be able to be used at real and practical sites for fluoride removal, especially in places that contain a slightly higher fluoride level.

### 3.4. The Stability of Adsorbents

When applying the PAs in a practical way for fluoride removal in groundwater, one of the greatest concerns is the leaching of pyrolusite components, which are highly conductive manganese and iron compounds. To meet the Thailand groundwater standard, a stability experiment was conducted by stirring both PA-1 and PA-2 with DI water for an hour and measuring the leaching of manganese and iron. The results found that after one hour, the amount of iron that is released in a solution of washed minerals with PA-1 and PA-2 is equal to 0.27 and 0.49 mg/L, respectively. These values are determined by the concentration of iron in the ore. To meet the requirements of the standard, the maximum amount of iron that can be found in groundwater that can be consumed is 0.5 milligrams per liter. Similar trends exist with manganese; the amount of manganese that is released into the solution when the ore is washed with PA-1 and PA-2 is, respectively, 0.14 mg/L and 0.22 mg/L. Following the manganese guidelines (not exceeding 0.5 mg/L), the maximum amount of manganese that can be present in groundwater that can be consumed is 0.3 mg/L. As a result, pyrolusite’s ability to absorb fluoride ensures that the amount of manganese and iron released into the solution does not exceed the acceptable limit.

## 4. Conclusions

The application of abundant pyrolusite ore (PA-1) and the preparation of a simple acid modification on its surface (PA-2) were both explored for their potential to remove fluoride by adsorption. Because of the features of acid modification, it is possible to improve an active site on the surface of pyrolusite in order to attain a better adsorption capacity than the one that was initially formed. The adsorption behavior represented not only physio-adsorption but also chemisorption, which was presented as an important mechanism due to the results of the fitted pseudo-second order kinetic and Freundlich isotherm models in addition to the monolayer adsorption. This was because the adsorption behavior represented both physical adsorption and chemical adsorption. In addition, the stability of the materials, including the value of manganese and iron, which are the primary components of pyrolusite ore, was safe for human consumption. This was the case despite the materials also being suitable for practical use. The application of a prepared low-cost material in this research can be employed for subsequent fluoride applications in large-scale and column tests, particularly in regions with an excessive concentration of fluoride.

## Figures and Tables

**Figure 1 ijerph-19-17103-f001:**
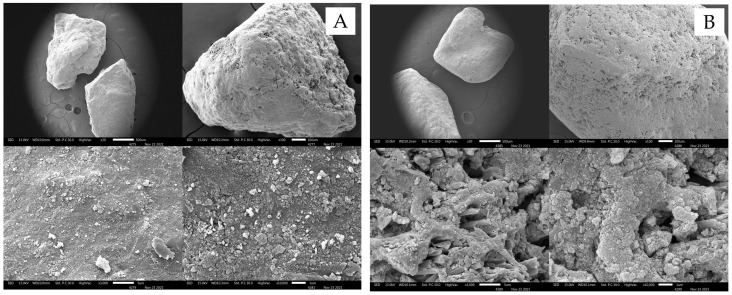
SEM analysis of (**A**) pyrolusite (washed by DI; PA-1) and (**B**) acid-modified pyrolusite (soaked with HCl; PA-2).

**Figure 2 ijerph-19-17103-f002:**
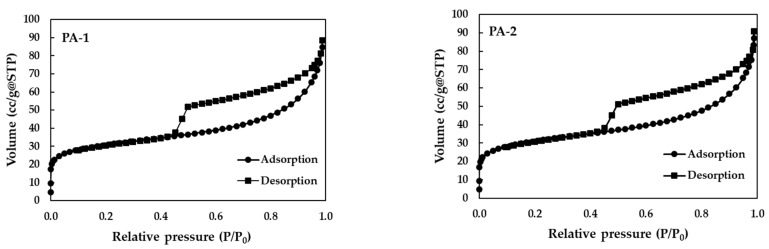
Nitrogen adsorption isotherm (BET) at 77 K of PA-1 and PA-2. (The samples were pre-heating at 200 °C for 16 h under vacuum conditions before measurement).

**Figure 3 ijerph-19-17103-f003:**
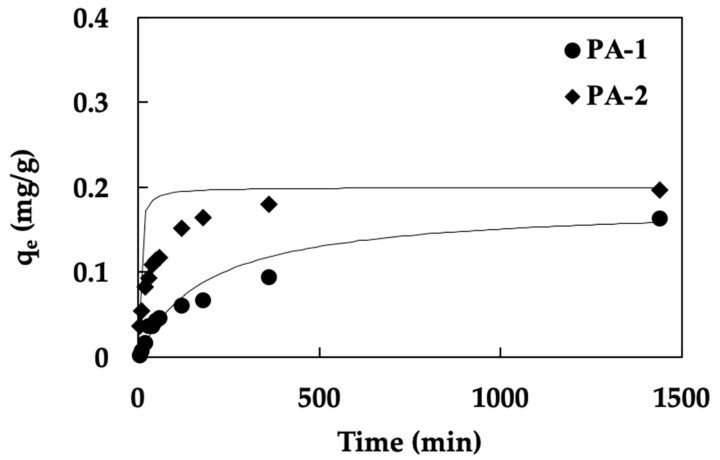
Adsorption kinetics of fluoride on PA-1 and PA-2 (pH 7, solid/liquid ratio 15 g/L, temperature 25 °C, vertical shaker at 200 rpm for 0.24 h). Solid lines exhibited the obtained data from the pseudo-second order kinetic model).

**Figure 4 ijerph-19-17103-f004:**
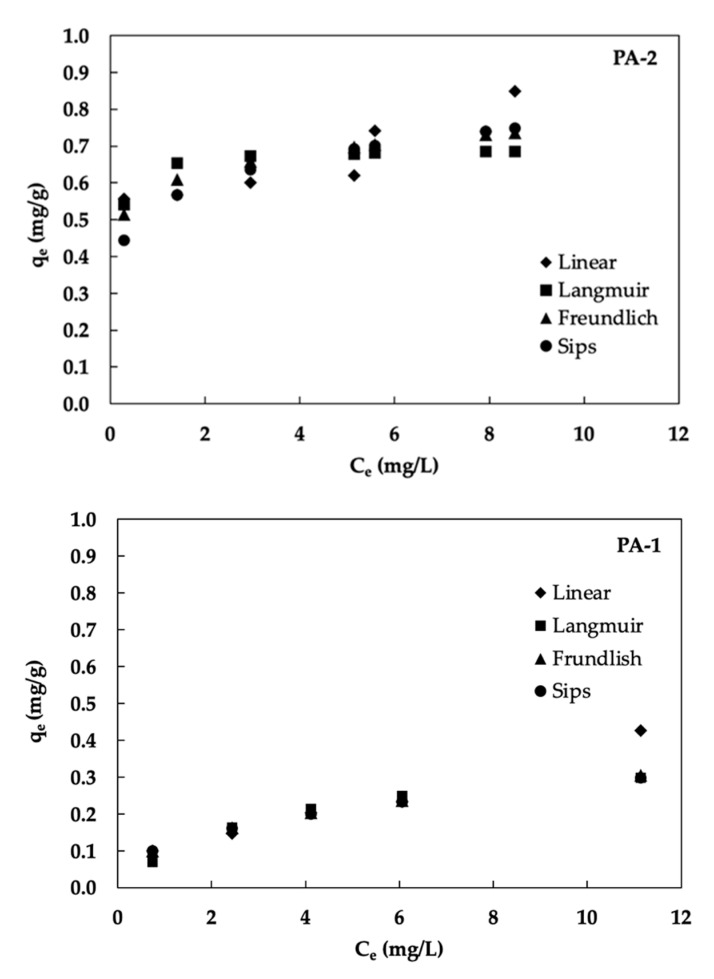
Adsorption isotherm of fluoride on PA-1 and PA-2 (pH 7, solid/liquid ratio 15 g/L, temperature 25 °C, and vertical shaker at 200 rpm at equilibrium time).

**Table 1 ijerph-19-17103-t001:** PA preparation and acid modification method.

Types of Modification		Modifier/Solution	Time	Temperature (°C)
Original-PA	PA-1	Deionized water (DI)	Soaked and washed with DI for 5 day	Room temperature
Acid-PA	PA-2	0.1 N HCl	Soaked with 0.1 N HCl for 24 h and wash with DI	Room temperature

**Table 2 ijerph-19-17103-t002:** Elemental analysis of PA materials by EDS mapping.

Materials	Elements (% wt)							Total
	C	N	O	Al	Si	Mn	Fe	
PA-1	18.09	4.47	19.01	7.87	1.87	28.20	20.49	100
PA-2	15.76	n.d. *	20.03	4.18	n.d.	23.77	36.26	100

* n.d. = not detected.

**Table 3 ijerph-19-17103-t003:** BET characterization results of the PA-1 and PA-2.

Materials	S_BET_ ^a^ (m^2^/g)	Pore Diameter ^b^ (nm)	Pore Volume ^c^ (cc/g)	pH_pzc_
PA-1	111.48	3.65	0.112	4.00
PA-2	108.93	4.62	0.107	4.00

^a^ BET surface area was conducted at a gas temperature of 77 K. ^b^ Adsorption average pore diameter ^c^ Pore volume analyzed by BJH model

**Table 4 ijerph-19-17103-t004:** Kinetic parameter of fluoride by pseudo-first order and pseudo-second order model.

Materials	q_e,exp_ (mg/g)	Pseudo-First Order			Pseudo-Second Order				
		K_1_ (1/min)	q_e_,_cal_ (mg/g)	R^2^	k_2_ (g/mg min)	q_e,cal_ (mg/g)	h (mg/g min)	t _1/2_ (min)	R^2^
PA-1	0.17	−0.0015	0.44	0.7951	0.03	0.18	0.0009	185.26	0.9575
PA-2	0.20	−0.0088	0.42	0.9860	0.15	0.20	0.0062	32.04	0.9995

**Table 5 ijerph-19-17103-t005:** Isotherm parameters of fluoride adsorption on PA-1 and PA-2 adsorbents.

Materials	Linear Isotherm		Langmuir Isotherm			Freundlich Isotherm			Sip Isotherm			
	*K*_p_ (L/g)	*R^2^*	q_m_ (mg/g)	K_L_ (L/mg)	*R^2^*	K_F_ (mg/g)	1n	*R^2^*	$q_{ms}$	*K_s_* (L/mg)	$n_{s}$	*R^2^*
PA-1	0.03	0.8939	0.38	0.62	0.8765	0.11	0.51	0.9488	50.07	0.002	2.47	0.9358
PA-2	0.03	0.8459	0.70	12.31	0.3605	0.58	0.56	0.9899	87.08	0.006	6.44	0.8821

**Table 6 ijerph-19-17103-t006:** Comparison of the differences between the prepared material and the reported literature regarding fluoride removal.

Materials	Adsorption Capacity (mg/g)	Operating Condition	References
*Polygonum orientale* Linn.	0.39 (at 20 °C)	C_Initial_ = 20 mg/L (pH = 7) Absorbance dose = 2 g/100 mL	[45]
*Polygonum orientale* Linn.—modification with Al_2_(SO_4_)_3_	0.77 (at 20 °C)	C_Initial_ = 20 mg/L (pH = 7) Absorbance dose = 2 g/100 mL	[45]
Iron ore	1.45 (at 22 °C)	C_Initial_ = 10 mg/L (pH = 6) Absorbance dose = 0.5 g/100 mL	[24]
Dolomite	0.011 (at room temp)	C_Initial_ = 10 mg/L (pH = 7) Absorbance dose = 10 g/100 mL	[15]
Pyrolusite (PA-1)	0.11 (at room temp)	C_Initial_ = 10 mg/L (pH = 7) Absorbance dose = 1.5 g/100 mL	This study
Acid-modified pyrolusite (PA-2)	0.58 (at room temp)	C_Initial_ = 10 mg/L (pH = 7) Absorbance dose = 1.5 g/100 mL	This study

## Data Availability

Not applicable.

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
