# Peer review of "Adsorption of Fluoride onto Acid-Modified Low-Cost Pyrolusite Ore: Adsorption Characteristics and Efficiencies"

_ijerph, 2022, doi:10.3390/ijerph192417103_

Round 1

Reviewer 1 Report

1)      The English language should be polished further by correcting any grammar mistakes.

2)      “…consuming too much of it…” (line 43). How much?

3)      Please change “… and Drug Administration of Thailand all agree…” (line 51)

to

“… and Drug Administration of Thailand, all agree…”

4)      References are missing in the phrases:

“… due to its low cost, ease of design and operation, and high level of effectiveness” (line 61)

“It was a good opportunity to study those pyrolusite ore, which was a low-cost material for fluoride removal.” (line 76)

“Even though some materials, such as iron ore, demonstrated a good performance, pyrolusite is a one abundant ore option from northern region of Thailand that will be able to be used on the real and practical sites for fluoride removal especially in the place that contains the slightly exceed of fluoride level.”

5)      “The porosity and surface area (BET) were measured by nitrogen adsorption isotherm using Autosorb 1 MP, Quantachrome Instrument, USA.” (line 108) At 77K?

6)      “…Barrett-Joyner-Halenda (BJH) methods…” (line 173)

7)      “The adsorption reached equilibrium after 60 min in all adsorbents which can be indicated a fast rate of the reaction and similar trend with other ore adsorbent”. Equilibrium or stationary state?

8)      All the equations mentioned in the article aren’t related to any reference.

9)      The authors should consider whether to rewrite the results and discussion section (compare the performance of the selected materials with that exhibited by other materials in terms of efficiency and costs - Table 6).

10)  Table 6 – Which were the operating conditions (pH, dose, initial concentration…)? Were the same operating conditions?

Author Response

To reviewer, please see the response to comment in the attached file.

Reviewer 2 Report

1.      The argumentation in the paper lacks basic understanding of the reaction. Some recent comprehensive papers including this area in the introduction should benefit readers'' understanding of this field as follows: International Journal of Hydrogen Energy 44 (2019), 16387-16399; International Journal of Hydrogen Energy 45 (2020), 15086-15099

2.      Why is HCl activation done? Structural changes after acid activation should be shown.

3.      Experimental data for operational parameters must be explained in Figure Captions
3. Further data on the substances used should be included (e.g. degree of purity, chemical formulas, etc.).
4. The author never says a single word about a number of the measurement repetitions neither he discusses standard deviations of the measured data. Some of the results are doubtful, without a thorough analysis of the measurement error; the credibility of the obtained results is considerably diminished.
6. What is the novelty that authors want to bring with respect to this issue? It should clearly include the purpose of the study in the last paragraph of the introduction

7. Which functional groups were attached to the surface after modification? One of the elemental analysis, FTIR or XPS analysis should be done.

Author Response

(The authors gave the same response as above.)

Reviewer 3 Report

The paper by Phacharapol Induvesa et al. entitled “A study of the adsorption characteristics and efficiencies of fluoride onto acid-modified low-cost pyrolusite ore” reports the utilization of the adsorbent pyrolusite, which is a manganese mineral largely composed of MnO2, was a major component. Lab-scale experiments were carried out to investigate the efficacy of original pyrolusite ore (PA-1) and acid-modification-PA (PA-2) created as low-cost adsorption materials for fluoride removal. According to the results of adsorption tests, PA that had been treated with an acid solution had the capacity to contain more fluoride than the PA that had been used initially. This was applicable in terms of both the adsorption rate and the performance at the equilibrium time. According to the findings of an isotherm, the primary adsorption behavior is determined by the effect that surface components and chemical composition have on porous materials. This is the first current study that provides a comparison of pyrolusite from Thailand's mining and basic modified pyrolusite to remove a fluoride contaminant in synthetic groundwater by an adsorption process. It will be able to be applied in the future to protect the exceeding fluoride concentration in household and drinking water treatment technology in the community. The study is novel and could be acceptable to IJERPH[IS1]  after addressing following minor comments:

·         Title of the manuscript requires significant improvement.

·         Keywords must be written in Alphabetical order.

·         Some quantitative details of the results to be added in the Abstract.

·         The results of Figure 2. Nitrogen adsorption isotherm (BET) results of PA-1 and PA-2, are not smoother, could the authors repeat the surface area measurement related test?

·         Several formatting issues are observed throughout the manuscript.

·         Authors have provided the Supplementary file. On the other hand, the manuscript states on Line 389: Supplementary Materials: Not applicable. Please remove this discrepancy.

·         Some relevant literature must be updated such as Environmental Science and Pollution Research 28, 9050–9066     

Author Response

(The authors gave the same response as above.)

Round 2

Reviewer 1 Report

Nothing to mention